# Elevated Serum Gastrin Is Associated with Melanoma Progression: Putative Role in Increased Migration and Invasion of Melanoma Cells

**DOI:** 10.3390/ijms242316851

**Published:** 2023-11-28

**Authors:** Akos Janos Varga, Istvan Balazs Nemeth, Lajos Kemeny, Janos Varga, Laszlo Tiszlavicz, Dinesh Kumar, Steven Dodd, Alec W. M. Simpson, Tunde Buknicz, Rob Beynon, Deborah Simpson, Tibor Krenacs, Graham J. Dockray, Andrea Varro

**Affiliations:** 1Department of Cellular and Molecular Physiology, Institute of Translational Medicine, University of Liverpool, Liverpool L69 7BE, UKg.j.dockray@liverpool.ac.uk (G.J.D.); avarro@liverpool.ac.uk (A.V.); 2Department of Dermatology and Allergology, University of Szeged, 6720 Szeged, Hungary; 3Department of Pathology, University of Szeged, 6725 Szeged, Hungary; 4Department of Biochemistry, Institute of Integrative Biology, University of Liverpool, Liverpool L69 7BE, UK; 5Department of Pathology and Experimental Cancer Research, Semmelweis University, 1085 Budapest, Hungary

**Keywords:** gastrin, CCK2R, CCKBR, melanoma, invasion, migration, secretome, MMP-2, TIMP-3

## Abstract

Micro-environmental factors, including stromal and immune cells, cytokines, and circulating hormones are well recognized to determine cancer progression. Melanoma cell growth was recently shown to be suppressed by cholecystokinin/gastrin (CCK) receptor antagonists, and our preliminary data suggested that melanoma patients with *Helicobacter* gastritis (which is associated with elevated serum gastrin) might have an increased risk of cancer progression. Therefore, in the present study, we examined how gastrin may act on melanoma cells. In 89 melanoma patients, we found a statistically significant association between circulating gastrin concentrations and melanoma thickness and metastasis, which are known risk factors of melanoma progression and prognosis. Immunocytochemistry using a validated antibody confirmed weak to moderate CCK2R expression in both primary malignant melanoma cells and the melanoma cell lines SK-MEL-2 and G361. Furthermore, among the 219 tumors in the Skin Cutaneous Melanoma TCGA Pan-Cancer dataset showing gastrin receptor (CCKBR) expression, significantly higher CCKBR mRNA levels were linked to stage III–IV than stage I–II melanomas. In both cell lines, gastrin increased intracellular calcium levels and stimulated cell migration and invasion through mechanisms inhibited by a CCK2 receptor antagonist. Proteomic studies identified increased MMP-2 and reduced TIMP-3 levels in response to gastrin that were likely to contribute to the increased migration of both cell lines. However, the effects of gastrin on tumor cell invasion were relatively weak in the presence of the extracellular matrix. Nevertheless, dermal fibroblasts/myofibroblasts, known also to express CCK2R, increased gastrin-induced cancer cell invasion. Our data suggest that in a subset of melanoma patients, an elevated serum gastrin concentration is a risk factor for melanoma tumor progression, and that gastrin may act on both melanoma and adjacent stromal cells through CCK2 receptors to promote mechanisms of tumor migration and invasion.

## 1. Introduction

Acquisition of mutations is a defining feature of cancer progression. Even so, it is clear that the micro-environment of mutation-bearing cells also plays a pivotal role [1]. The relevant micro-environmental factors include extracellular matrix, stromal and immune cells, and cytokines and circulating hormones.

The gene encoding the stomach hormone gastrin is normally expressed in gastric endocrine cells, and the main active products are C-terminally amidated peptides of 17 (G17) or 34 amino acid residues [2] that have similar affinity for the gastrin receptor, known as the cholecystokinin-2 receptor (CCK2R; formerly the gastrin-CCKb receptor, encoded by the *CCKBR* gene). Physiological interactions between gastrin and CCK2R stimulate gastric acid secretion and increase gastric epithelial cell proliferation, maturation, and migration.

Several tumor types express the gastrin gene (with the exception of neuroendocrine tumors); however, post-translational processing in these cells generally stops short of generating amidated gastrins, and the main products (progastrin and G17 or G34 extended by C-terminal Gly) are not thought to be physiological ligands for CCK2R. The latter is, however, expressed by esophageal, gastric, and pancreatic tumor cells, and circulating amidated gastrin may influence tumor progression [3,4]. Moreover, in transgenic mice with elevated serum gastrin, *Helicobacter-*induced gastric cancer is accelerated [5]. Gastrin induces migration and invasion in cancer cell lines, at least in part, by activating extracellular proteases of the matrix metalloproteinase (MMP) family [6].

Melanoma arising from transformed melanocytes is one of the most severe types of skin cancer, with a significant risk of metastasis [7]. In rodent models, endothelial-derived cancer-associated fibroblasts (CAF) promote intravasation and dissemination of melanoma cells [8]. Signaling can occur through paracrine-acting cytokines, chemokines, or other mediators such as tumor-derived exosomes [9,10]. Melanoma progression from the radial to vertical phases requires cancer cells to penetrate the basement membrane. Exposure of melanoma cells to an altered microenvironment leads to genomic and proteomic alterations thought to be important for adapting to the new environment [11,12,13,14,15,16].

There is some evidence that melanoma cells express CCK2R [17], suggesting that gastrin may act on these cells. Furthermore, melanoma cell growth was recently shown to be suppressed by cholecystokinin/gastrin (CCK) receptor antagonists [18]. In the present study, we have explored this hypothesis through re-examination of CCK2R expression, determination of circulating amidated gastrin in melanoma patients, and exploration of the effects of gastrin on melanoma cell migration and invasion.

## 2. Results

### 2.1. Melanoma Cells Express Functional CCK2 Receptors

A previously well-characterized CCK2R antibody [19,20] revealed some level of CCK2R expression by tumor cells in 15 out of the 18 MM patients compared with 2 out of the 10 BCC patients used as controls (Figure 1B). In the majority (56%) of the MM patients, the percentage of melanoma cells that exhibited CCK2R expression ranged from 15% to 20%, while in the expression in the remaining cases varied between undetectable (17%) and homogenous strong positivity (27%). 

Immunolabelling for simultaneous detection of CCK2R and Melan-A (a widely recognized melanoma marker) revealed a low percentage (2–3%) of normal epidermal melanocytes that expressed the CCK2 receptor. Densitometry of double-labelled samples revealed abundant Melan-A positive cells, but consistently low CCK2R expression in BCC compared to MM (Mann–Whitney U-test, *p* < 0.05; cytoplasm of 5 cells/field) (Figure 1B). Independent of the tumor type, there were stromal cells of fibroblastic origin (i.e., dermal fibroblasts, myofibroblasts) that expressed CCK2R, consistent with previous finding that myofibroblasts provide a novel pool of CCK2R-expressing cells [20].

Two representative melanoma cell lines, SK-MEL-2 and G-361, were explored as putative models for the actions of gastrin (see Section 4 for details for the main features of these cells). Peroxidase and immunofluorescence staining of the cultured cells revealed CCK2R expression in melanoma cells (16.6 ± 2.8 and 24.7 ± 1.9%, SK-MEL-2 and G361 respectively) (Figure 1A). Moreover, using qPCR, we identified CCK2R transcripts in both cell lines. Using GAPDH as a reference, the ΔCt values for SK-MEL-2 and G-361 were 15.9 ± 0.2 and 15.2 ± 0.2, respectively. Gastrin stimulation of CCK2R in many cell types is associated with increased intracellular calcium. The concentrations of intracellular Ca^2+^ in unstimulated melanoma cells were generally steady, but upon addition of G17 (10 nM), prompt and sustained increases in cytosolic calcium were observed, specifically in a subset of cells (6.6 ± 1.1 and 7.9 ± 1.8 % of G-361 and SK-MEL-2 cells, respectively; n = 3 independent experiments with recordings from a total of 246 and 204 cells, respectively) (Figure 1C).

### 2.2. Serum Gastrin Is Elevated in Advanced Melanoma

We then examined serum gastrin concentrations in 89 melanomas with a mean tumor thickness of 2.9 ± 0.3 mm. There were 45 patients with stage 1 (pT1a, pT1b, pT2a) and 44 with stage 2 (pT2b, pT3 and pT4) disease according to the latest AJCC classification at the time of writing. The patients were allocated to groups based on the information available from the primary histology. Interestingly, the probability of elevated gastrin was significantly greater in the sub-group of melanoma patients with stage 2 (18 out of 44) compared with stage 1 (3 out of 42) disease (OR 8.5, *p* < 0.0005; Fisher exact test *p* < 0.0003) (Figure 2A). Compatible with this, significantly elevated concentrations of serum gastrin were observed in melanoma patients belonging to the stage 2 group when compared to those in the stage 1 group (*t*-test *p* < 0.001; 40 ± 6.8 vs. 24 ± 4.4 pM, respectively), as well as when compared to 26 BCC patients as controls. 

Melanoma patients who had been prescribed acid inhibitors showed a higher likelihood of experiencing hypergastrinemia, as indicated by an odds ratio of 6.7 (*p* < 0.001; Fisher exact test *p* < 0.0011) (Figure 2B). There were also moderately raised fasting serum gastrin concentrations associated with *H. pylori* sero-positive melanoma patients (OR 5.36, *p* < 0.0023; Fisher exact test *p* < 0.002). Both the effect of acid inhibitors and of *H. pylori* infection are in accordance with the literature [21,22]. All but two of the melanoma patients with serum gastrin above the reference range were either *H. pylori positive* or on proton pump inhibitor (PPI) medication, or both. Moreover, restaging of patients following the acquisition of lymph node data from sentinel biopsy and/or lymph node dissection showed a correlation between elevated serum gastrin concentration and melanoma progression, as indicated by the presence of regional lymph node metastasis (Figure 2C). In line with the protein data, elevated CCKBR gene expression was statistically linked to stage III–IV melanomas compared to early stage I–II melanomas when examining the CCKBR-expressing subset of the melanoma TCGA PanCancer dataset (PMID: 29625048) (Figure 2D). 

### 2.3. CCK2R Activation by Gastrin Stimulates Melanoma Cell Migration and Invasion

Initial experiments confirmed the absence of a proliferative effect of gastrin on melanoma cells [17] (See Appendix A). In contrast, gastrin significantly increased migration and invasion in the Boyden chamber experiments (one-way ANOVA, *p* < 0.01). However, these effects were suppressed when melanoma cells were treated with the CCK2R antagonist, L740093 (one-way ANOVA, *p* < 0.01) (Figure 3).

### 2.4. Secretome Analysis Reveals Specific Targets of Gastrin

A secretome analysis using a modified SILAC technique was applied to melanocytes to identify the putative targets of gastrin [23]. In the SK-MEL-2 and G-361 cell media, 1266 and 1507 proteins were identified, respectively, of which 134 and 187 were labelled. Of these, approximately 50% corresponded to secretory proteins based on the SigP criteria (listed in Appendix A). 

The set of proteins identified as classically secreted using SignalP v. 4.1 was further analyzed using Panther v 10.0 for the identification of the potential pathways differentially influenced by gastrin [24]. An over-representation analysis conducted on both cell lines demonstrated that gastrin-induced upregulation of classically secreted proteins primarily affected the biological function related to cellular adhesion, including cell–cell adhesion and cell–matrix adhesion. This observation was validated by adhesion assays, where gastrin stimulated a significant, concentration-dependent increase in the number of adherent cells in both cell lines (control vs. 10nM G17: student t-test, *p* = 0.025 and 0.019 in SK-MEL-2 and G-316, respectively).

Gastrin-regulated secretory proteins were found to exhibit distinct protein class characteristics. In G-361 cells, these proteins were identified as metalloprotease inhibitors, whereas in SK-MEL-2 cells, they were characterized as proteases. Specifically, an elevation in MMP-2 secretion (H to L ratio of 1.5) was observed in SK-MEL-2 cells. Conversely, G-361 cells exhibited a significant decrease in the secretion of the metalloproteinase inhibitor TIMP-3, and to a lesser extent, TIMP-1 and TIMP-2 (with H to L ratios of 0.125, 0.87, and 0.7, respectively). These findings were confirmed through Western blotting in gastrin-treated cells, demonstrating a significant increase in MMP-2 levels in SK-MEL-2 cells and a significant reduction in the expression levels of TIMP-3 and TIMP-1, respectively, in G-361 cells (n = 3 for each). 

Gastrin did not exert any significant effect on MMP-2 secretion in G-361 cells or TIMP-3 inhibition in SK-MEL-2 cells (Figure 4). ELISA assays further confirmed previous findings, showing elevated MMP-2 concentrations in SK-MEL-2 cells (student t-test, *p* = 0.029), while a decrease in TIMP-3 secretion was observed in G-361 cells upon exposure to gastrin (student *t*-test, *p* = 0.015).

### 2.5. MMP-2 siRNA Transfected SK-MEL-2 Cells Exhibit Decreased Migration and Invasion in Response to Gastrin

In order to examine the functional implications of the observed alterations in MMP-2 secretion described above, siRNA was employed to suppress MMP-2 expression in SK-MEL-2 cells. The stimulatory effect of gastrin on the migration of these cells in Boyden chambers was significantly impaired after MMP-2 knockdown (gastrin-treated: control vs. MMP-2 knock down; one-way ANOVA; *p* < 0.05) (Figure 5). The invasive capacity of SK-MEL-2 cells in Boyden chambers was significantly compromised upon downregulation of MMP-2 (gastrin-treated: control vs. MMP-2 knock down; one-way ANOVA, *p* < 0.05) (Figure 5). 

### 2.6. TIMP-3 Inhibits Gastrin-Stimulated Migration in G-361 Cells

In G361 cells, gastrin treatment led to reduced secretion of TIMP-3, as indicated by the secretome analysis. Subsequently, the addition of TIMP-3 resulted in the reduction of migrating cells in the Boyden chambers compared to the control group (TIMP-3 w/G17 vs. hG17; one-way ANOVA; *p* < 0.05) (Figure 6). The observed inhibitory effect was also evident in Boyden chamber invasion assays (TIMP-3 w/G17 vs. G17; one-way ANOVA; *p* < 0.05) (Figure 6).

### 2.7. Influence of the Microenvironment on the Migratory and Invasive Behavior of Gastrin-Stimulated Melanoma Cells 

We next examined the invasion in a more complex system in which melanoma cells were incubated in the presence of extracellular matrix and a key stromal cell type, fibroblasts. Since CCK2R can be expressed by some cells of fibroblastic lineage [20], we first examined its expression in dermal fibroblasts and myofibroblasts. Notably, a distinct subset of skin stromal cells exhibited expression of CCK2R, with dermal fibroblasts at a proportion of 12.7 ± 2.1% and myofibroblasts at 11.1 ± 2.3%.

When spheroid cultures of SK-MEL-2 cells were embedded in a Matrigel–collagen matrix supplemented with dermal fibroblasts, a 1.3 ± 0.1-fold increase in the average surface was observed over a period of 6 days, compared to control spheroids that were devoid of stromal cells in their vicinity (Kruskal–Wallis test; fibroblasts in matrix vs. matrix alone, *p* = 0.01) (Figure 7A). The impact of dermal fibroblasts on cancer cell proliferation and expansion, as determined by the spheroid surface area, was not influenced by gastrin (Kruskal–Wallis test; fibroblasts w/ G17 vs. fibroblasts only, *p* = 1.0) (Figure 7A), consistent with the observation that gastrin had no effect on cell growth. 

However, individual cancer cells invade the matrix in this system, and this was stimulated by dermal fibroblasts in combination with gastrin compared to either gastrin or fibroblasts alone (one-way ANOVA, G17 vs. G17 w/fibroblasts, *p* < 0.001; fibroblasts vs. G17 w/fibroblasts, *p* < 0.001; w/G17 vs. w/o G17, *p* = 0.023) (Appendix A).

In a second model more closely resembling the organization of human skin, SK-MEL-2 cells were seeded on top of type I collagen/Matrigel containing dermal stromal cells. In this setup, the interface between the cancer cells and stromal cells exhibited a similar structure to the junction between the epidermis and dermis. The integrity of this border was maintained in the control samples throughout the entire observation period of two weeks. When melanoma cells were layered on ECM with dispersed fibroblasts, clusters of cancer cells, characterized by pedicles or detached aggregates (as shown in Figure 7B), exhibited the capability to invade and infiltrate into deeper layers. The invasion was therefore clearly stimulated by the presence of dermal fibroblasts (one-way ANOVA, ECM w/fibroblasts vs. ECM w/o cellular components, *p* < 0.001). In the absence of dermal cells, gastrin only slightly increased the invasion (one-way ANOVA, ECM w/G17 vs. ECM w/o G17, *p* < 0.043). However, the invasion of tumor cells stimulated by the presence of fibroblasts was increased further by treatment with gastrin (one-way ANOVA, ECM w/ fibroblasts and G17 vs. ECM w/o fibroblasts and G17, *p* < 0.001; ECM w/fibroblasts vs. ECM w/fibroblasts and hG17, *p* = 0.002) (Figure 7B). The data therefore point to indirect effects of gastrin in stimulating melanoma cell invasion mediated by dermal fibroblasts.

## 3. Discussion

Expression of CCK2R in the brain and gut is physiological [2], but it is also expressed in cancer [25,26]. The present study supports previous work suggesting that CCK2R is expressed in melanoma cells [17] and takes this observation further by showing (a) an association between elevated serum gastrin and melanoma patients with more severe disease, and (b) gastrin stimulation of the invasion of melanoma cells through mechanisms involving modulation of MMP2/TIMP secretion. Crucially, however, these direct effects appear to be relatively weak, and in the presence of matrix and dermal fibroblasts, gastrin-stimulated melanoma cell invasion is indirect and is mediated by fibroblasts, at least some of which also express CCK2R. Taken as a whole, the data raise the possibility that increased circulating gastrin concentration should be considered a modifiable risk factor for melanoma progression.

Epidemiological data reveal a substantial, 60% escalation in the occurrence of MM among Caucasian populations over the preceding three decades [7,27]. Here, we demonstrated both at the mRNA and protein levels that a proportion of skin melanomas show a consistent yet heterogeneous expression of CCK2R in both cancerous cells and the adjacent stromal environment. However, it is important to note that this study did not include an assessment of alternative splice variants of the CCK2R in the examined melanoma specimens, which could be a valuable area for future investigation [28,29]. Nevertheless, together, the data suggest that CCK2R should now be considered a candidate for mediating the hormonal effects of cutaneous melanoma.

The proportions of melanoma patients with hypergastrinemia attributable to *H. pylori* or acid-inhibitor therapy were similar to those in a control group (BCC) and were in the same range reported for other populations [30,31]. Surprisingly, higher fasting serum gastrin concentrations were associated with advanced melanoma stages. We do not propose that melanoma patients are more likely than any other group to exhibit increased serum gastrin. However, the data suggest that in those melanoma patients with increased serum gastrin, the hormone may play a role in disease progression. Long-term utilization of acid-secretory inhibitors, *H. pylori* infection, or both may therefore pose increased risks for melanoma progression. Conversely, it is now appropriate to ask whether treatments that would lower circulating gastrin in the relevant patient group (withdrawal of acid inhibitor drugs, eradication of *H. pylori*) would be beneficial.

In two melanoma cell lines (SK-MEL-2 and G-361), we confirmed that gastrin did not increase cell proliferation (Appendix A) [17]. Inevitably, cell lines do not fully represent the complexity of a tumor; however, they are important and useful model systems which have made a major contribution to melanoma research since their introduction in the late 1960s [32]. Several studies have assessed and compared the characteristics of melanoma tissue samples and cancer cell lines [33,34]. In this study, we selected SK-MEL-2 and G361 cell lines, as they have been confirmed to be transcriptional representatives of their tumorous counterparts [33]. It is noteworthy that somatic mutations and copy number alterations generally occur at similar rates in melanoma patient samples and melanoma cell lines, with the exception of BRAF and TP53, which exhibit higher mutation frequencies. Therefore, we incorporated a BRAF-negative (SK-MEL-2) cell line. Briefly, SK-MEL-2 is a microsatellite-instable/MSI high melanocytic cell line with mutations in NRAS, TERT, and TP53, but with a wild-type BRAF status [35]. On the other hand, G-361 is a BRAF/V600E and TERT mutant amelanocytic melanoma cell line [36]. Our findings indicated that gastrin increased intracellular Ca^2+^ and robustly increased migration and invasion of both cell lines through mechanisms sensitive to a CCK2R antagonist, providing functional and pharmacological evidence for CCK2R expression. The secretomes derived from melanoma cells were subjected to proteomic analysis, leading to the identification of candidate proteins that are responsible for the rapid degradation of the extracellular matrix in response to gastrin stimulation.

Matrix metalloproteinases have been extensively studied as pivotal factors involved in the remodeling of the extracellular matrix, facilitating local tumor growth and the infiltration of cancer cells into vascular and lymphatic vessels [37,38]. Moreover, upregulation of MMP-2 has been described in multiple highly aggressive melanoma cell lines [13,39]. We found that gastrin upregulated MMP2 and downregulated TIMP3 expression. The application of siRNA to inhibit MMP-2 expression or introduction of TIMP-3 partially reversed the effects of gastrin. Thus, the proteomic and functional data suggest that gastrin influences melanoma progression by targeting components involved in the regulation of protease activity and inhibitor function. This ultimately leads to accelerated extracellular matrix (ECM) degradation, thereby promoting tumor spread.

The action of gastrin on cell invasion in the ECM was substantially enhanced by dermal fibroblasts. Since stromal cells also express functional CCK2R, gastrin could act both directly to stimulate melanoma cells and indirectly via stromal cells. We suggest that gastrin liberates factors from stromal cells (i.e. members of the EGF and IGF families have already been established as gastrin targets in other systems [40]) which then stimulate melanoma invasion and migration. The underlying mechanisms remain to be investigated in melanoma. Even so, our observations highlight the significance of the tumor microenvironment and bidirectional signaling between epithelial and stromal cells in influencing melanoma progression.

## 4. Materials and Methods

### 4.1. Patients

Serum gastrin was measured in malignant melanoma (MM) patients (*n* = 89, 39 M) or a control group of basal cell carcinoma (BCC) patients (*n* = 26, 16 M) (Appendix A). All diagnoses were verified by histology of biopsies. Serum samples were obtained for radioimmunoassays (RIAs) and serology. Anonymized patient data was processed in accordance with EU General Data Protection Regulation laws. A second group of patients was used for studies of stromal cell influences on melanoma cell responses to gastrin. In this case, dermal fibroblasts were cultured from resected skin samples of patients, where surgery was performed due to a suspected skin malignancy (dysplastic nevi, BCC, squamous cell cancer), confirmed by histology. This study was conducted in line with the approval granted by the National Ethical Board (TUKEB; MCC-INTER-002, No.: IF-964-2/2016).

### 4.2. Cells

Melanoma cells (G-361 and SK-MEL-2) were obtained from the American Type Culture Collection (ATCC, Manassas, VA, USA) and cultured in RPMI-1640 (Sigma-Aldrich Poole, Dorset, UK) supplemented with 10% FBS and 1% antibiotic/antimycotic solution. Detailed genetic information on the G-361 (RRID:CVCL_1220) and SK-MEL-2 (RRID:CVCL_0069) cell lines is available at https://www.cellosaurus.org/ (accessed on 2 October 2023). Fibroblasts were cultivated in fibroblast basal medium (Lonza, Cambridge, UK) supplemented with 1% penicillin/streptomycin and 1% glutamine (Sigma Aldrich, Poole, Dorset, UK) and utilized within 10 passages. To induce differentiation into dermal myofibroblasts, the cells were incubated in culture medium containing 10 ng/mL of transforming growth factor beta 1 (TGF-β1) for a duration of 72 h [41].

### 4.3. Immunohistochemistry

Formal-fixed paraffin-embedded sections were processed for automated immunohistochemistry using a Leica BOND Max Autostainer with antibodies against CCK2R (polyclonal rabbit, HPA041976, Sigma, diluted 1:200, HIER pH9 for 60 min) and Melan-A (clone A103, Dako, dilution of 1:300, HIER pH9 for 20 min). Double labelling was accomplished using the Chromoplex Dual Detection Kit. Receptor expression was assessed semi-quantitatively, based on the intensity of the reaction (0; 1+; 2+; 3+).

### 4.4. Immunocytochemistry

Cultured cells were formalin-fixed (4% *w*/*v*, permeabilised with 0.2% Triton X-100 in PBS, PBS-T) for 30 min at ambient temperature and processed for immunohistochemistry using a CCK2R antibody (Atlas AB, Bromma, Sweden) according to the manufacturer’s instructions and an appropriate fluorescein-labelled secondary antibody raised in donkey (Jackson Immunoresearch, Soham, UK). Alternatively, in some cases, the melanoma cells were stained with peroxidase using the ABC staining system (ThermoFisher, MA, USA).

### 4.5. TCGA Database Analysis and pPCR of Melanomas for CCKBR Expresssion

*CCKBR* gene expression was examined in the Skin Cutaneous Melanoma TCGA Pan-Cancer dataset (PMID: 29625048), involving 443 melanoma patients, accessed from cBioPortal (PMIDs: 22588877, 23550210, 37668528) (https://www.cbioportal.org/study/summary?id=skcm_tcga_pan_can_atlas_2018, accessed on 28 September 2023). After removing the samples with no CCKBR expression (i.e., RSEM value = 0), a Wilcoxon test was used to compare the Z-scored CCKBR transcript abundance in patient cohorts of disease stages I–II (n = 120) to those found in stage III–IV melanomas (n = 99). The data cleaning was performed in R v4.2.0, and the ggplot2, ggbeeswarm, and ggpubr R packages were used for plotting and statistics.

Furthermore, RNA extracted from the cancer cells was reverse transcribed using Promega AMV RT reagents [19]. Multiplex real-time qPCR was performed using an Applied Biosystems AB7500 system with manufacturer-supplied software ver2.3. The following Taqman primer/probe pairs were employed: GAPDH: 5′-GCT CCT CCT GTT CGA CAG TCA-3′(forward), 5′-ACC TTC CCC ATG GTG TCT GA-3′ (reverse), 5′-CGT CGC CAG CCG AGC CAC A-3′ (probe); CCK2R: 5′-TGA CTC TGG GAT GCT CCT AGT-3′ (forward), 5′-TGG TCA GAG GTA TGA GAT TAG GC-3′ (reverse), 5′-ACC TCA CAG TGA CCC TTC CCA ATC AGC-3′ (probe). The results were expressed as ∆CT normalized to GAPDH.

### 4.6. Radioimmunoassay and ELISA

Serum gastrin concentrations were quantified using a radioimmunoassay (RIA) method, employing an antibody specific to the C-terminus of G17. This antibody exhibits equal reactivity towards both G17 and G34, but does not bind progastrin or C-terminal variants of G17. The *H. pylori* status was determined using serology [19].

### 4.7. Calcium Imaging

Melanoma cells were seeded at a density of 1−3 × 10^4^ cells per well in 6-well plates, incubated 24 h, then loaded with 2 μM Fluo-4 AM (Invitrogen, Paisley, Renfrewshire, UK) and stimulated with G17 (10 nM) using ionomycin (1 µM) as a positive control [20].

### 4.8. Flow Cytometry

Cultured cells were collected and fixed with 4% paraformaldehyde (PFA) at 37 °C for 10 min. Subsequently, permeabilization was performed using 90% methanol at 4 °C for 30 min. The cells were incubated with 4′,6-diamidino-2-phenylindole dihydrochloride (DAPI) at a concentration of 1 µg/mL. Finally, cell sorting was carried out using a FACS Canto II flow cytometer according to a previously described protocol [20].

### 4.9. Cell Migration and Invasion Assays

Transwell migration and invasion assays were performed using BD inserts (Corning, New York, NY, USA) (25,000 cells per insert). Human G17 (10 nM) (Bachem, St Helens, Merseyside, UK) was added to the lower well, either with or without the CCK2R antagonist, L-740093 (50 µM, kindly provided by Dr R. Freidinger, Merck Sharpe and Dohme, Rathway, NJ, USA), or recombinant human tissue inhibitor of matrix metalloproteinase (TIMP)-3 or matrix metalloproteinase (MMP)-2 (R&D Systems, Minneapolis, MN, USA), as appropriate.

### 4.10. Spheroids

SK-MEL-2 cells were seeded at a density of 3000 cells per droplet and incubated for up to five days, either in the presence or absence of myofibroblasts at a concentration of 10,000 cells per well [42]. The growth of spheroids was determined by calculating the change in surface area relative to day 0 and expressed as a fold-change. Additionally, the invasion of individual melanoma cells was evaluated by measuring the distance of cancer cell migration from the center of the spheroid at four different angles. The experiments were conducted in triplicate, and for each spheroid, four fields were examined and evaluated. 

### 4.11. Organotypic Cultures

Following a previously described protocol [43], SK-MEL-2 cells at a concentration of 1 × 10^6^ were placed on a layer consisting of a 1:1 mixture of Matrigel (Corning, NY, USA) and collagen-I (Millipore, MA, USA). This setup was performed both in the presence and absence of myofibroblasts at a concentration of 0.5 × 10^6^ cells. On day 3, the culture was raised onto a wire gauze and maintained with an air-medium interface for 21 days, changing the media every 48 h. Invasion was determined as the depth of invading cancer cells into the Matrigel/collagen layer using the contact point of epithelial cells and Matrigel/collagen as a baseline. The experiments were performed in triplicate and two sections were scored, giving a total of 6 observations from each organoid.

### 4.12. Proteomic Analysis

Putative gastrin targets in the secretomes of SK-MEL-2 and G-361 cells were identified using a modified SILAC protocol that exploits the rapid synthesis of secreted proteins compared to cellular proteins [23]. Briefly, the cells were incubated with G17 for 24 h, and for the last 6 h, medium containing either 12C6 lysine (light label), or 13C6 lysine (heavy label) was added to label secreted proteins. StrataClean resin (Agilent Technologies Ltd., Wokingham, UK) was used to capture proteins in the media samples prior to analysis using a Nano-Acquity (Waters) reverse phase HPLC system in-line with an LTQ Orbitrap Velos (Thermo). The SILAC data were searched and analyzed using MaxQuant 1.1.1.36 against the human IPI database v3.68 using the recommended default settings.

### 4.13. Secreted Protein Search and GeneOntology Analsysis

Data filtered after FDR 1% for heavy to light ratio peptides with a cut-off 0.01 were imported to the Uniprot database to generate fasta sequence files. To identify high score signal peptides among the list of experimentally obtained proteins, classical (with a D cut-off greater than 0.5) and non-classical (with an NN score cut-off greater than 0.6) secreted proteins were identified using SignalIP v.4.0 and SecretomeP v.2.1. The dataset generated from the analysis of secreted proteins containing signal peptides was uploaded to Panther v.10. software to predict protein classes, molecular functions, and biological processes that showed significant enrichment (*p* < 0.05).

### 4.14. Western Blotting

Culture media was concentrated using StrataClean resin (Agilent Technologies Ltd., Santa Clara, CA, USA) and prepared for Western blotting. Antibodies against MMP-2, TIMP-3 (R&D Systems Abingdon, UK; AF902 and MAB973, respectively), TIMP-1, TIMP-2, prosaposin, and MMP-1 (R&D Systems, Abingdon, UK; AF980, AF971, AF8520, and MAB901, respectively) were utilized. A densitometric analysis of the immunoblots was performed using the Molecular Imaging Software 4.1 of Kodak Image Station 4000 MM (Kodak, Rochester, NY, USA) and Image J 1.48v. The results represent the mean of three independent experiments. 

### 4.15. MMP-2 Knockdown

The Amaxa™ Cell line Nucleofector™ kit V with the T-19 program (Amaxa, Köln, Germany) was used for temporary transfection of cells to achieve MMP-2 knockdown. The transfection procedure was carried out in accordance with the manufacturer’s instructions. Melanoma cells were then treated with 3 µM of validated siRNA (Sigma-Aldrich Poole, Dorset, UK) specific to MMP-2 (AM16708, Life Technologies, Warrington, UK). The effectiveness of the MMP-2 knockdown was confirmed through Western blot analysis.

### 4.16. Statistics 

Data are presented as mean ± standard error of means (SEM). The statistical analyses involved parametric tests such as student t-test, Fisher exact test, Pearson’s chi-squared test, and ANOVA, as appropriate. In certain instances, non-parametric tests such as the Mann–Whitney U test and Kruskal–Wallis test were utilized. Statistical significance was determined at *p* < 0.05. The software packages used for the analyses were Systat Software Inc. v. 12.0 (London, UK) and IBM SPSS Statistics V26.0 (New York, NY, USA).

## 5. Conclusions

The main findings of this study are that heterogenous gastrin receptor (CCK2R) expression can be detected both in cultured and clinical melanoma samples, and that elevated serum gastrin concentrations exhibit an association with melanoma thickness; a well-known risk factor of melanoma progression. 

Altogether, our findings suggest that gastrin may exert both direct and indirect effects on melanoma cells by enhancing their migration and invasion capabilities, possibly by altering the balance towards the dominance of secreted proteases, e.g., MMP-2, against protease inhibitors, e.g., TIMP-3. These effects are likely to be mediated through CCK2 receptors expressed by dermal fibroblasts, myofibroblasts, and to a lesser extent, by cancer cells. Moreover, the release of other paracrine-acting growth factors (IGFI-II) and chemokines (specifically, HB-EGF) from the tumor stroma may further facilitate cancer progression [38,44,45]. The accurate mechanism of action may require further molecular and functional studies [20,40]. The data raise the prospect that circulating gastrin is a risk factor for melanoma progression. 

## Figures and Tables

**Figure 1 ijms-24-16851-f001:**
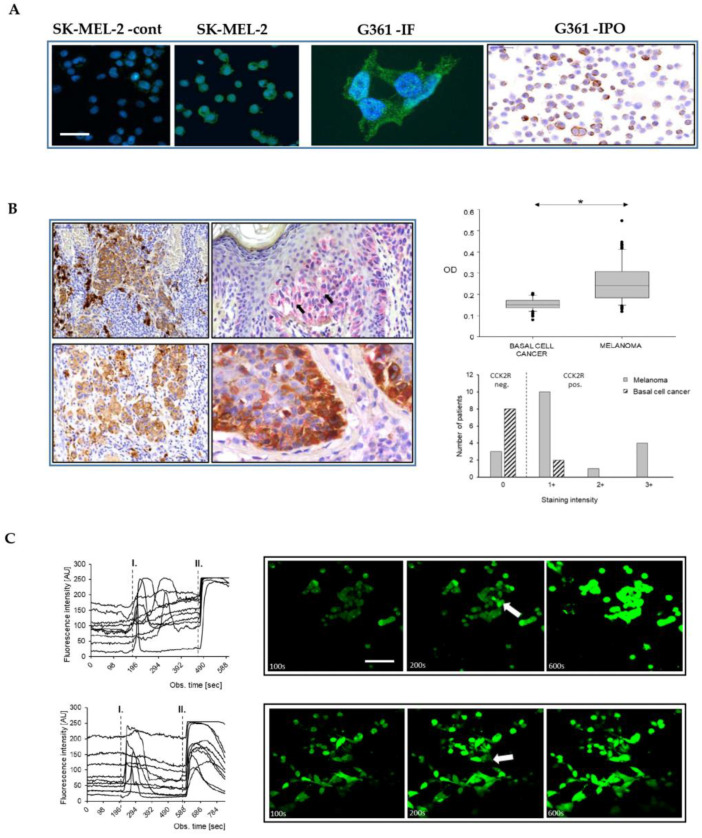
Melanoma cells express functional CCK2 receptors. Moderate CCK2R immunofluorescence in melanoma cell lines SK-MEL-2 and G361, respectively, beside a negative control (SK-MEL2 –cont). CCK2R expression confirmed in cytospined G361 cells using immunoperoxidase staining (G361-IPO) (**A**). Scale bar 30 µm in all images, except for high power G361 (10 µm). Both immunoperoxidase (brown) and immunoalkaline phosphatase reactions (pink) (counterstained with hematoxylin) (**B**, left panel) demonstrate CCK2R positive nodular, superficial spreading (upper left and right), as well as invasive nodular and lentigo malignant (lower left and right) melanomas in the tissue sections (arrows indicate CCK2R positive cells). The latter double labeling shows overlapping Melan A (brown) and CCK2R (pink) immunoreactions. Please note that the strong brown cells on upper left are melanophages. Quantitative analysis of the optical density of CCK2R staining in tissue sections of BCC and MM demonstrated higher values in the MM samples, (**B** right panel) (* *p* < 0.05). Gastrin-stimulated melanoma cells (SK-MEL-2 upper, G-361 lower panel **C**, scale bar 30 µm) showed a transient increase in fluorescence of Fluo-4 AM (Invitrogen, Paisley, UK) as a result of CCK2R activation (mark I). As a positive control, ionomycin (mark II) was used. Representative images captured at specific time points illustrate the basal fluorescence (left, 100 s), the effect of gastrin treatment (middle, with arrows indicating responder cells, 200 s), and the ionomycin control (right panel, 600 s).

**Figure 2 ijms-24-16851-f002:**
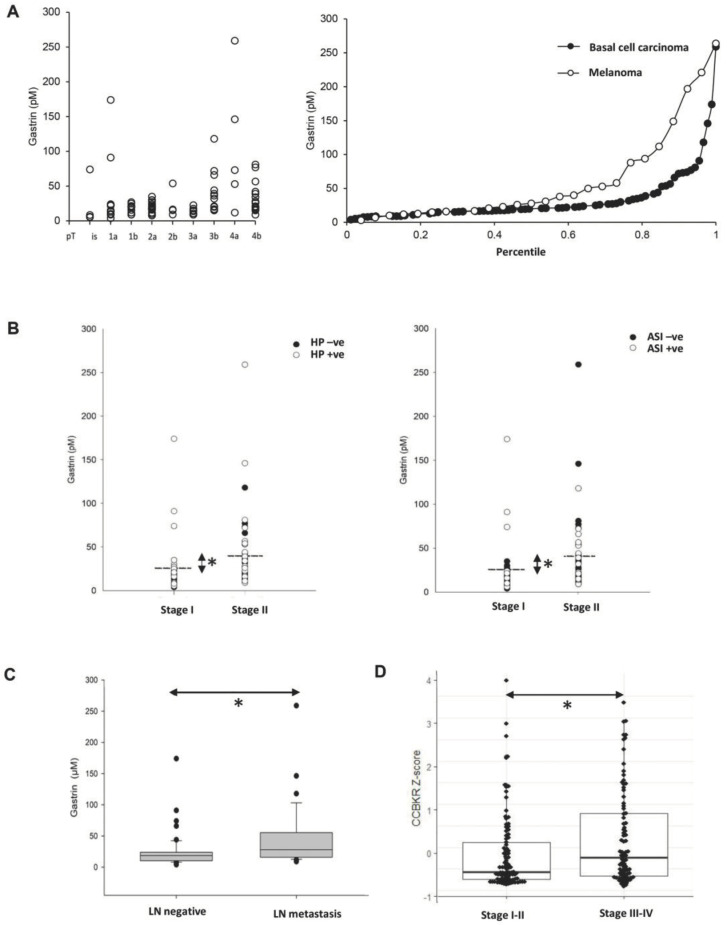
Expression of gastrin in clinical melanoma samples. The serum gastrin concentration was elevated in melanoma patients with advancing disease stages (**A** left) and compared to basal cell carcinoma patients (**A** right). The serum gastrin was also significantly higher (Mann–Whitney Rank Sum Test, * *p* < 0.01) in stage 2 (pT > 2b) than in stage 1 (pT < 2a) melanoma patients both in the *H. pylori*-positive (HP + ve, **B** left) vs. negative or in the acid inhibitor therapy-receiving (ASI + ve, **B** right) vs. untreated cohorts. Dashed lines indicate mean of serum gastrin concentrations in stage 2 (40 ± 6.8 pM) vs. stage 1 (24 ± 4.4 pM) cases. Increased serum gastrin concentrations were also associated with regional lymph node metastasis (* *p* = 0.015) (**C**). CCKBR mRNA levels were also significantly higher in stages III–IV compared to the stage I–II melanoma cohort (Wilcoxon * *p* = 0.012) in the TCGA datasets (PMID: 29625048) as tested using the cBioPortal (**D**).

**Figure 3 ijms-24-16851-f003:**
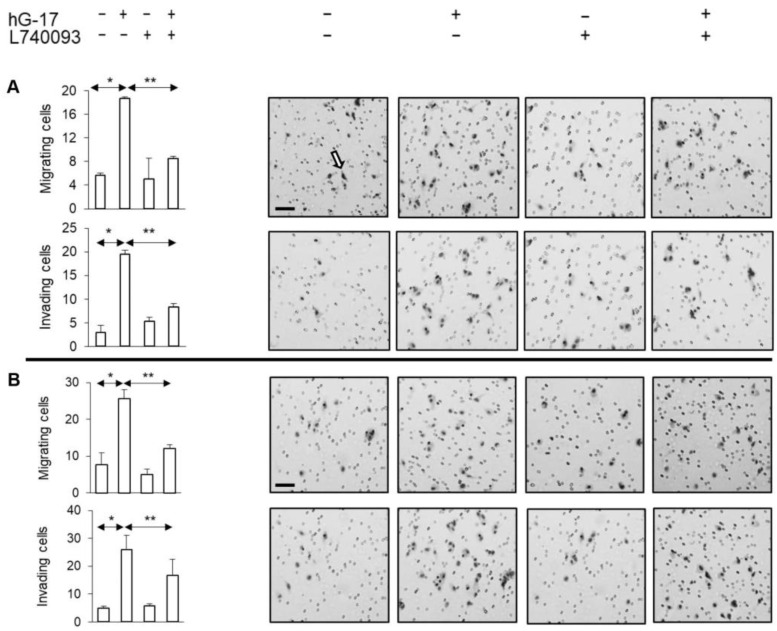
Gastrin stimulated migration and invasion of melanoma cells (**A** SK-MEL-2; **B** G-361). Right panels display representative images of the membranes acquired from Boyden inserts and Matrigel Biocoat chambers. Arrow indicates methylene blue-stained cells. (Scale bar 50 µm in all images. *, ** refer to *p* < 0.01 and <0.005, respectively).

**Figure 4 ijms-24-16851-f004:**
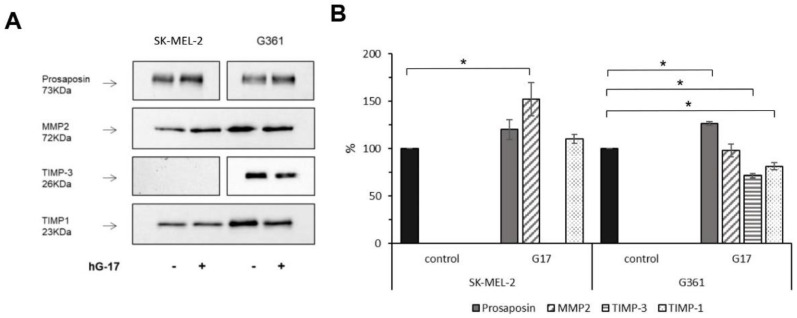
Conditioned media from G17-stimulated melanoma cells exhibit upregulation of MMP-2 and downregulation of TIMP-3 expression. Representative demonstration of the confirmation of proteomic data using Western blot analysis, with prosaposin serving as a benchmark for gastrin responsiveness (**A**). Mean and standard deviation (±SD) of the densitometric analysis results from three independent experiments (**B**). Statistical difference (* *p* < 0.05) between treated and untreated control groups (considered as 100%) are indicated as follows: prosaposin, TIMP-3 and TIMP-1 levels in G361 cells and MMP2 expressions in the SK-MEL-2 cell line.

**Figure 5 ijms-24-16851-f005:**
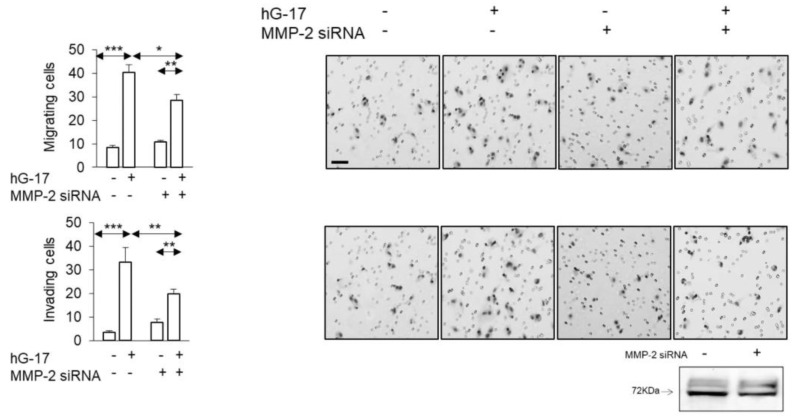
Gastrin-stimulated SK-MEL-2 melanoma cultures transfected with MMP-2 siRNA show suppressed migration and invasion. (Scale bar, 50 µm in all images. *, ** and *** refer to *p* < 0.01, 0.005 and <0.0001, respectively.) Western blot validated the successful transfection of MMP-2 siRNA.

**Figure 6 ijms-24-16851-f006:**
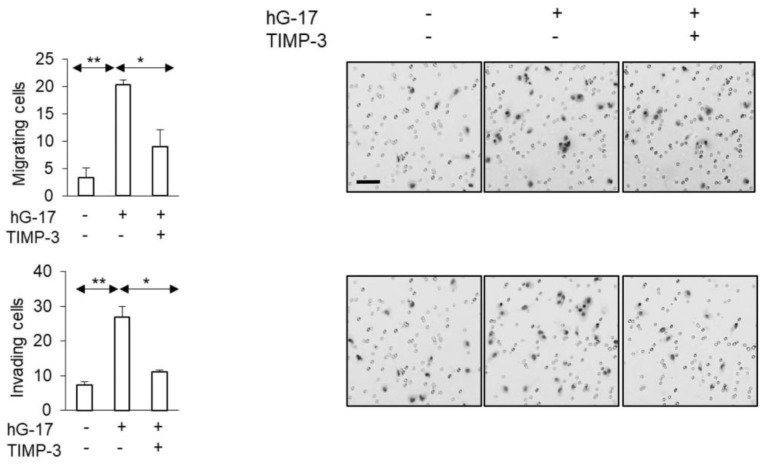
TMIP-3 supplementation of G-361 melanoma cell medium attenuated the impact of gastrin on the migratory and invasive properties of cancer cells. (Scale bar 50 µm in all images. *, ** refer to *p* < 0.01 and <0.005, respectively).

**Figure 7 ijms-24-16851-f007:**
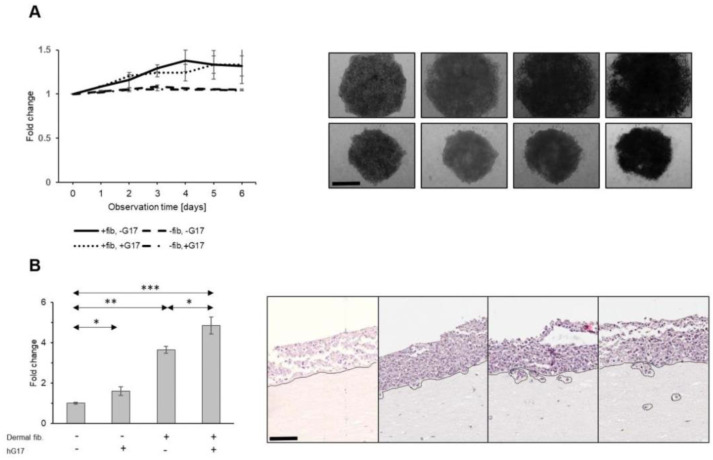
Dermal fibroblasts promote the growth of melanoma spheroids in a gastrin-independent manner (**A**). Representative images of melanoma spheroids embedded in acellular (lower) vs. collagen matrix with fibroblasts dispersed throughout (upper panel). Gastrin stimulates invasion of melanoma tumor cells in skin organoids (**B**) (depth of melanoma invasion measured from tumor–stromal interface, expressed as a fold change; scale bar 30 µm for **A** and 90 µm for **B**). *, **, *** refer to *p* < 0.01; <0.005 and <0.0001, respectively.

## Data Availability

Datasets related to this article can be found at: Varga, Akos (2022), “Proteomic analysis of melanoma secretome (SK-MEL-2, G361) after gastrin treatment”, Mendeley Data, V1, doi: 10.17632/xmnvxzkvx4.1.

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
