# Peer review of "Elevated Serum Gastrin Is Associated with Melanoma Progression: Putative Role in Increased Migration and Invasion of Melanoma Cells"

_ijms, 2023, doi:10.3390/ijms242316851_

Round 1

Reviewer 1 Report

Comments and Suggestions for Authors

The research focuses primarily on the microenvironmental factors that influence cancer progression, with a particular focus on melanoma. The authors investigate the relationship between melanoma growth and cholecystokinin/gastrin (CCK) receptor antagonists. They note that patients with Helicobacter gastritis, which is associated with elevated serum gastrin levels, may be at increased risk for cancer progression. The study also explores the role of gastrin in modulating melanoma cells, highlighting the correlation between circulating gastrin concentrations and specific melanoma characteristics. Notable findings include the identification of CCK2R receptor expression in melanoma cells and cell lines, the association of elevated CCKBR mRNA levels with metastatic melanoma, and the effect of gastrin on cell migration and invasion, highlighting the contributions of MMP-2 and TIMP-3. Overall, the manuscript is clear and systematically presents its content, ensuring a smooth flow and easy comprehension of all figures. However, to improve the quality of the manuscript, I have a few suggestions.

Based on my review:

Please include a scale bar in the figures. Some, such as Figure 1A, seem to lack this important visual element.

For consistency, I recommend keeping the same magnification in all parts of Figure 1A.

In Figure 1C, the name of the dye should be included in the figure legend. In addition, the time points should be indicated on each figure.

For Figure 1, it would be beneficial to include a Western blot (WB) to validate the antibody used.

In Figure 4A, I suggest ensuring that the Western blot data presented is consistent. Instead of showing segmented versions, run the samples together for a better presentation.

For Figure 5, it would be helpful to include a Western blot to verify the efficiency of the knockdown.

I believe these revisions will further refine and improve the quality of the manuscript.

Author Response

Dear Editor,

We are grateful for the reviewers’ suggestions. Please find below our point-by-point answers to their criticism and our modifications, additions in red color in the revised manuscript.

Answers to reviewer 1.

  1. We inserted scale bars into all relevant figures and information in their legends.
  2. We harmonized all figure components in Figure 1A to be the same magnification except the high power review image which demonstrates more details of the reaction.
  3. As suggested by the reviewer we specified the applied fluorophore in the figure legend also. Timestamps for representative images have also been added.
  4. The applied antibody used for immunocytochemical localization of CCK2R was extensively validated previously by comparing wild type gastric (AGS) and oesophageal adenocarcinoma (OE33) derived cancer cells (which do not express the receptor) with their counterparts that have been stably transfected with cDNA encoding CCK2R i.e. AGS-Gr and OE33-Gr cells. (Garalla HM et al. Physiol Rep. 2018;6(10):e13683. doi:10.14814/phy2.13683; Varga A et al. Physiol Rep. 2017;5(19):e13394. doi:10.14814/phy2.13394).
  5. Instead of using several raw Western Blot images we summarized their main messages in combined panel Figure 4A. Nevertheless, as it was requested, we are sending you the original blot images.
  6. In agreement with the reviewer’s suggestion, we added a Western blot to confirm successful transfection of MMP-2 siRNA.

Answer to reviewer2.

We thank the positive views on our manuscript by the 2nd reviewer and agree with him/her that further efforts are useful to get deeper insight into our findings, as we wrote in our conclusion:

"The accurate mechanism of action may require further molecular and functional stud-ies [27,35]."

We hope that our revised manuscript is suitable for publication in the special issue of “Biomarkers of tumor progression…” in IJMS.

Sincerely yours:

Tibor Krenacs

PhD, DSc

Reviewer 2 Report

Comments and Suggestions for Authors

The study tries to demonstrate an association between elevated serum gastrin levels and clinical progression of melanoma.

The authors found that melanomas expressed higher levels of cholecystokinin/gastrin 2 receptor (CCK2R) than controls in patients with NMSC. In particular, stage II melanomas have higher gastrin levels than stage I melanomas.

Biologically, it would appear that activation of CCK2R by gastrin does not so much promote proliferation of melanoma cells as it does their ability to invade and penetrate.

The article supports a very audacious and attractive thesis.

The steps of the experiments are adequately described.

English is well written.

Although further studies are needed to properly determine the pathogenesis of CCK-supported tumor growth, the article is of great interest.

Author Response

Dear Editor,

We are grateful for the reviewers’ suggestions. Please find below our point-by-point answers to their criticism and our modifications, additions in red color in the revised manuscript.

Answer to reviewer2.

We thank the positive views on our manuscript by the 2nd reviewer and agree with him/her that further efforts are useful to get deeper insight into our findings, as we wrote in our conclusion:

"The accurate mechanism of action may require further molecular and functional stud-ies [27,35]."

We hope that our revised manuscript is suitable for publication in the special issue of “Biomarkers of tumor progression…” in IJMS.

Sincerely yours:

Tibor Krenacs

PhD, DSc